# FOXP2 exhibits projection neuron class specific expression, but is not required for multiple aspects of cortical histogenesis

**Ryan J Kast[†]\*, Alexandra L Lanjewar[†], Colton D Smith, Pat Levitt\***

Department of Pediatrics and Program in Developmental Neuroscience and Neurogenetics, The Saban Research Institute, Children's Hospital Los Angeles, Keck School of Medicine, University of Southern California, Los Angeles, United States

**Abstract** The expression patterns of the transcription factor FOXP2 in the developing mammalian forebrain have been described, and some studies have tested the role of this protein in the development and function of specific forebrain circuits by diverse methods and in multiple species. Clinically, mutations in *FOXP2* are associated with severe developmental speech disturbances, and molecular studies indicate that impairment of *Foxp2* may lead to dysregulation of genes involved in forebrain histogenesis. Here, anatomical and molecular phenotypes of the cortical neuron populations that express FOXP2 were characterized in mice. Additionally, *Foxp2* was removed from the developing mouse cortex at different prenatal ages using two Cre-recombinase driver lines. Detailed molecular and circuit analyses were undertaken to identify potential disruptions of development. Surprisingly, the results demonstrate that *Foxp2* function is not required for many functions that it has been proposed to regulate, and therefore plays a more limited role in cortical development than previously thought.

DOI: https://doi.org/10.7554/eLife.42012.001

\*For correspondence:
rkast@mit.edu (RJK);
plevitt@med.usc.edu (PL)

[†]These authors contributed equally to this work

**Competing interests:** The authors declare that no competing interests exist.

## Introduction

The proper function of the cerebral cortex requires the formation of highly stereotyped circuits during development. These circuits are built through interdependent processes, including proliferation of neural progenitors, migration of neurons to their appropriate positions, morphological and physiological differentiation of diverse neuron subtypes, and the formation of synapses of requisite strength between appropriate pairs of neurons. Impairments in these fundamental aspects of development can lead to lifelong dysfunction of the cortex, which is believed to contribute to core symptoms of many neurodevelopmental disorders (*Rubenstein and Rakic, 2013*).

The winged helix transcription factor, FOXP2, has been implicated in ontogenetic processes relevant to the development of the cerebral cortex (*Vernes et al., 2011*; *Chiu et al., 2014*; *Chen et al., 2016*), and some studies have directly implicated FOXP2 in cortical ontogeny (*Tsui et al., 2013*; *Garcia-Calero et al., 2016*). FOXP2 expression in the developing cortex is restricted to subpopulations of post-mitotic neurons in the deep cortical layers, a pattern that is highly conserved across mammalian species (*Ferland et al., 2003*; *Takahashi et al., 2003*; *Campbell et al., 2009*; *Mukamel et al., 2011*). Mutations in *FOXP2* cause a severe developmental speech and language disorder, known as childhood apraxia of speech (*Lai et al., 2001*; *MacDermot et al., 2005*). Human neuroimaging and animal studies have identified alterations in basal ganglia function that could contribute to the clinical disorder symptoms (*Vargha-Khadem et al., 1998*; *Belton et al., 2003*; *Groszer et al., 2008*; *French et al., 2012*; *Chen et al., 2016*), but whether changes in cerebral cortical organization and function are critically involved in impairments associated with *FOXP2* mutations is currently unknown.

The present study aimed to establish a more detailed understanding of the cell-type identity of FOXP2[+] neurons in the developing murine cerebral cortex, using molecular and neuroanatomical phenotyping approaches. Further, this study applied gold-standard conditional mouse genetics to selectively remove *Foxp2* from the developing cerebral cortex at different prenatal ages to ascertain its putative roles in the normal histogenic processes that generate the canonical six layers, specific cell types based on gene expression, and basic axon targeting to subcortical structures. The results show that FOXP2 expression is limited to specific corticofugal neuron populations and suggest that the gene plays a more limited role in mouse corticogenesis than previously concluded based on results obtained by other experimental methodologies.

## Results

### Foxp2 expression is enriched in developing corticothalamic projection neurons

Initial *Foxp2* expression mapping studies determined that *Foxp2* transcript and protein expression begin prenatally, with the onset of protein expression delayed relative to the mRNA, and with protein present primarily in postmitotic neurons (*Ferland et al., 2003*). However, other more recent studies have reported that FOXP2 protein is also expressed in mitotic progenitor cells (*Tsui et al., 2013*), where it regulates cortical neurogenesis. FOXP2 immunohistochemistry of coronal sections of the embryonic forebrain suggested that FOXP2 protein expression begins between embryonic day (E) 14.5 and E16.5 within postmitotic neurons of the infragranular layers (*Figure 1A,B*). Postnatally, it is well established that *Foxp2* expression is limited to glutamatergic neurons of the infragranular cortical layers, with robust expression predominantly in layer 6 (*Ferland et al., 2003*; *Hisaoka et al., 2010*; *Sundberg et al., 2018*). Layer 6 contains many FOXP2[+] neurons, whereas layer 5 contains some FOXP2[+] neurons that are more abundant in medial cortical areas at early postnatal stages (*Ferland et al., 2003*; *Campbell et al., 2009*). Layer 6, where most of the FOXP2[+] neurons are located, is comprised of two primary glutamatergic cortical neuron populations, corticothalamic (CT) and corticocortical (CC) neurons (*Thomson, 2010*; *Petrof et al., 2012*; *Harris and Shepherd, 2015*). To determine whether FOXP2 expression is selective to one of these populations or is expressed in both layer 6 projection neuron subtypes during development, on postnatal day (P) 12, the neuroanatomical tracer cholera toxin subunit B (CTB) was injected into primary somatosensory thalamus or primary motor cortex in separate cohorts of mice. Cell bodies of CT or CC neurons residing in layer 6 of primary SSC were retrogradely labeled. Sections through primary SSC were then stained with an anti-FOXP2 antibody and cellular expression of FOXP2 was assessed among retrogradely labeled neurons ipsilateral to the tracer injections. FOXP2 expression was evident in most CT neurons (Mean ± SEM, 78.3 ± 2.9%), but was expressed by very few CC neurons (Mean ± SEM, 6.4 ± 1.7%; *Figure 1C–E*).

Next, FOXP2 expression was assessed among molecularly-defined CT and CC neurons at several postnatal developmental stages. The CT-specific cre-driver mouse, Ntsr1-cre, was crossed with a cre-dependent Rosa-tdTomato reporter line (*Ai14*) to selectively and comprehensively label layer 6 CT neurons with tdTomato. FOXP2 immunohistochemistry revealed that nearly all tdTomato-positive neurons in the primary SSC co-expressed FOXP2 at P0, P7, and P14 (*Figure 1F–G*, Mean ± SEM: P0 = 90 ± 2%, P7 = 87 ± 2%, P14 = 91 ± 1%). For further molecular characterization, we also examined FOXP2 overlap with the receptor tyrosine kinase MET, which is expressed in CC neurons of layer 6, but nearly excluded from layer 6 CT neurons in SSC (*Kast et al., 2019*). Using co-labeling in Met[GFP] reporter mice (*Kast et al., 2019*), analysis of FOXP2 expression among GFP[+] layer 6 CC neurons revealed relatively few double-positive neurons at P0, P7, and P14 (*Figure 1H–I*, Mean ± SEM: P0 = 19 ± 2%, P7 = 16 ± 2%, P14 = 5 ± 2%). This finding is consistent with the limited expression of FOXP2 by layer 6 CC neurons observed through retrograde tracing. It is noteworthy, however, that there are FOXP2 and Met[GFP] double-positive cells positioned in the subplate/layer 6B, similar to the observation of Ntsr1-cre and Met[GFP] colocalization in deep layer six previously reported (*Kast et al., 2019*). Notably, the FOXP2 and GFP double-labeled neurons became quite sparse by P14, perhaps due to the downregulation of GFP in the subplate or to the programmed cell death of some subplate neuron populations, as has been reported previously during early postnatal development (*Hoerder-Suabedissen and Molnár, 2013*). We also examined two other layer 6 neuronal subtype

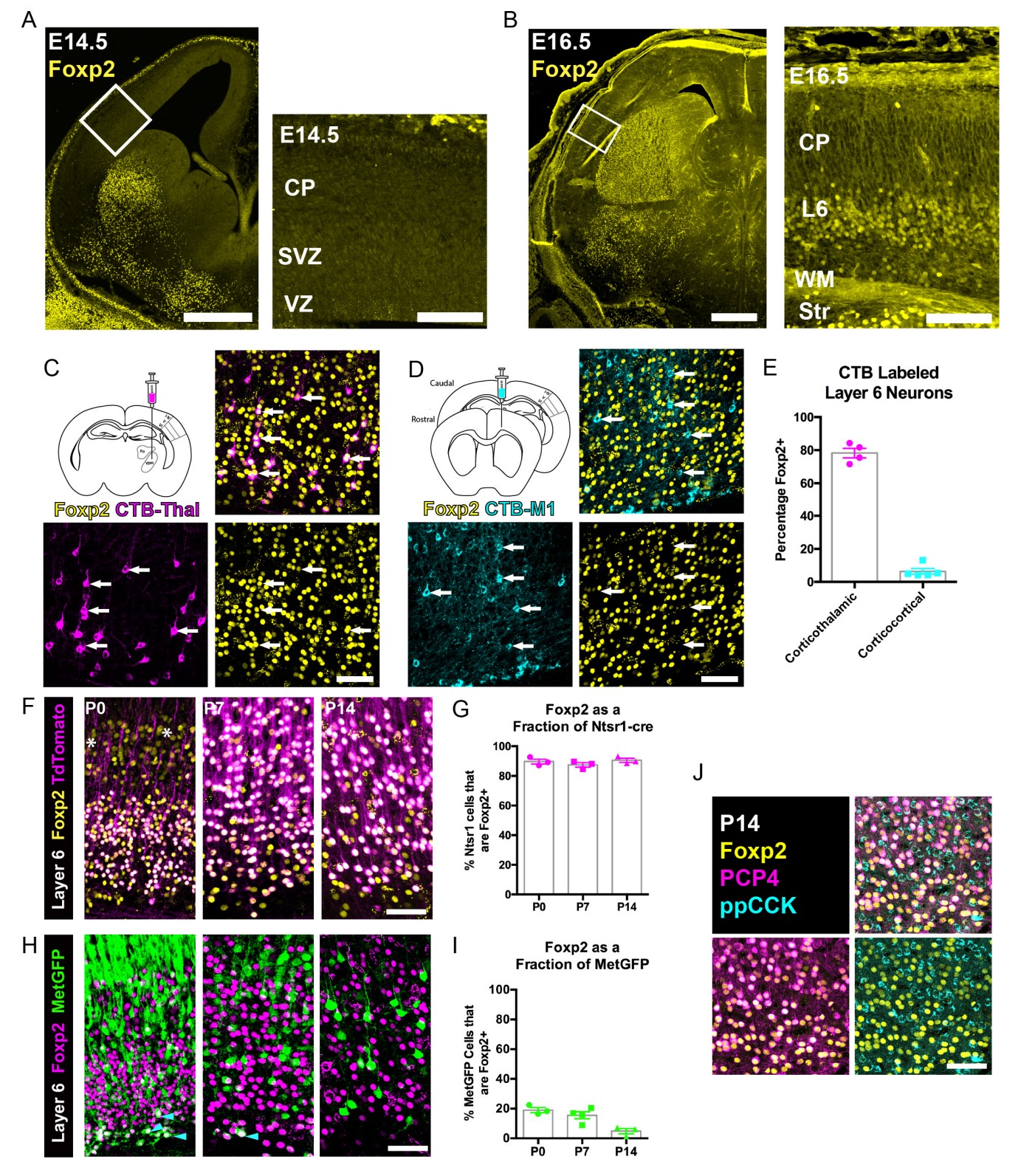

**Figure 1.** FOXP2 is enriched in corticothalamic neurons during cortical development. (**A**) Low magnification (left) and high magnification (right) images of FOXP2 (yellow) immunohistochemical labeling of E14.5 forebrain reveals absence of expression in the dorsal pallium, whereas the developing striatum is robustly labeled at this timepoint (N = 5). (**B**) Images of FOXP2 immunolabeling at E16.5 demonstrates the presence of FOXP2 expression within the deep layers of the developing cortical plate (N = 3). (**C**) Retrograde labeling of layer 6 corticothalamic neurons (magenta) by injection of CTB

*Figure 1 continued on next page*

*Figure 1 continued*

into the ventrobasal thalamus, combined with FOXP2 (yellow) immunohistochemistry at P14. (D) Corticocortical neurons (cyan) labeled by injection of CTB into the ipsilateral primary motor cortex, combined with FOXP2 (yellow) immunohistochemistry at P14. White arrows denote retrogradely labeled projection neurons. (E) Quantification of the percentages of retrogradely labeled corticothalamic (N = 4 mice) and corticocortical (N = 5) neurons that express FOXP2. (F) FOXP2 (yellow) immunohistochemistry in sections of P0, P7, and P14 somatosensory cortex from *Ntsr1-cre; tdTomato* mice (tdTomato is magenta); white asterisks denote relatively low-level expression in layer 5 at P0. (G) Quantification of the percentages of tdTomato-positive neurons that express FOXP2 at each age (P0, N = 3; P7, N = 3; P14, N = 3). (H) FOXP2 (magenta) immunohistochemistry in sections of P0, P7, and P14 somatosensory cortex from Met$^{GFP}$ (green) mice. Cyan arrowheads denote sparse FOXP2$^+$ and GFP$^+$ double-labeled cells localized to layer 6B/subplate. (I) Quantification of the percentages of GFP+ neurons that co-express Foxp2 at each age (P0, N = 3; P7, N = 4; P14, N = 3). (J) FOXP2 (yellow) colocalizes with PCP4 (magenta), but not ppCCK (cyan) (N = 3). Scale bars: 500 µm, A, B low magnification; 100 µm A, B high magnification; 50 µm C, D, F, H and J.

DOI: https://doi.org/10.7554/eLife.42012.002

The following source data and figure supplements are available for figure 1:

**Source data 1.** Foxp2 expression among retrogradely labeled and molecularly defined developing layer 6 neuron classes.
DOI: https://doi.org/10.7554/eLife.42012.005

**Figure supplement 1.** FOXP2 is transiently expressed by a subpopulation of PT neurons.
DOI: https://doi.org/10.7554/eLife.42012.003

**Figure supplement 1—source data 1.** Foxp2 expression among retrogradely and molecularly defined developing pyramidal tract neurons.
DOI: https://doi.org/10.7554/eLife.42012.004

marker genes, PCP4 and ppCCK (*Watakabe et al., 2012*). There was extensive colocalization between FOXP2 and PCP4, a marker of corticothalamic neurons, in layer 6, but there was minimal colocalization between FOXP2 and ppCCK, a marker of CC neurons (*Figure 1J*). Together, the molecular and connectivity analysis of FOXP2+ neurons in layer six indicate that FOXP2 expression is highly enriched in CT neurons of primary SSC during postnatal development. This is consistent with analysis in primary visual cortex of adult mice, which showed that *Foxp2*/FOXP2 expression is nearly exclusive to Ntsr1-cre expressing (CT) neurons in the adult (*Tasic et al., 2016*; *Sundberg et al., 2018*). The present study builds on previous findings by demonstrating minimal colocalization with two layer 6 CC markers (MET and CCK). Notably, lower level expression of *Foxp2* transcript has been detected in molecularly-defined subcerebral projection neurons by RNA-sequencing at perinatal stages (*Molyneaux et al., 2015*), consistent with the low-level expression observed in layer 5 at P0 (*Figure 1F*). Colocalization analysis with the pyramidal-tract (PT) neuron marker gene CTIP2 (*Arlotta et al., 2005*), confirmed that these layer 5 neurons are PT-type neurons (*Figure 1—figure supplement 1*). However, in contrast to the temporally stable and relatively uniform expression of FOXP2 by CT neurons in layer 6 of the postnatal SSC (*Figure 1*), FOXP2 was expressed by a minor subset of CTIP2$^+$ PT neurons at P0 (34 ± 3%) and P7 (10 ± 2%) (*Figure 1—figure supplement 1A–C*), and was almost completely absent from retrogradely labeled PT neurons in Layer 5 of SSC by P14 (4.4 ± 1.7%) (*Figure 1—figure supplement 1D,E*).

## FOXP2 is not required for normal histogenesis of the cerebral cortex

Given the enrichment of FOXP2 in corticofugal neuron subclasses in SSC, coupled with previous reports of fundamental roles for *Foxp2* in cortical neuron development (*Clovis et al., 2012*; *Tsui et al., 2013*; *Garcia-Calero et al., 2016*), we next used a direct genetic deletion strategy to examine putative involvement of FOXP2 in the development of anatomical and molecular properties of CT neurons. For these studies, mice harboring a *Foxp2* conditional allele (*Foxp2$^{fx}$*) were bred with the CT-specific reporter line *Ntsr1-cre; Rosa-tdTomato* (*Bortone et al., 2014*; *Kim et al., 2014*). Immunohistochemistry verified the elimination of FOXP2 from tdTomato+ neurons of *Ntsr1-cre; Foxp2$^{fx/fx}$* mice (*Figure 2A*, inset). tdTomato-expressing cell bodies remained limited to layer 6 of the cerebral cortex across genotypes, consistent with the absence of gross changes in laminar patterns due to *Foxp2* deletion. Confocal microscopy further revealed no overt changes in the distribution of tdTomato-labeled neurites in more superficial layers of cortex, suggesting minimal morphological rearrangement of the CT neuron population in response to *Foxp2* deletion. Inspection of tdTomato-labeled efferent axons arising from the deep layer neurons revealed a nearly identical pattern of CT innervation across *Foxp2* genotypes (*Figure 2A*). There was normal fasciculation

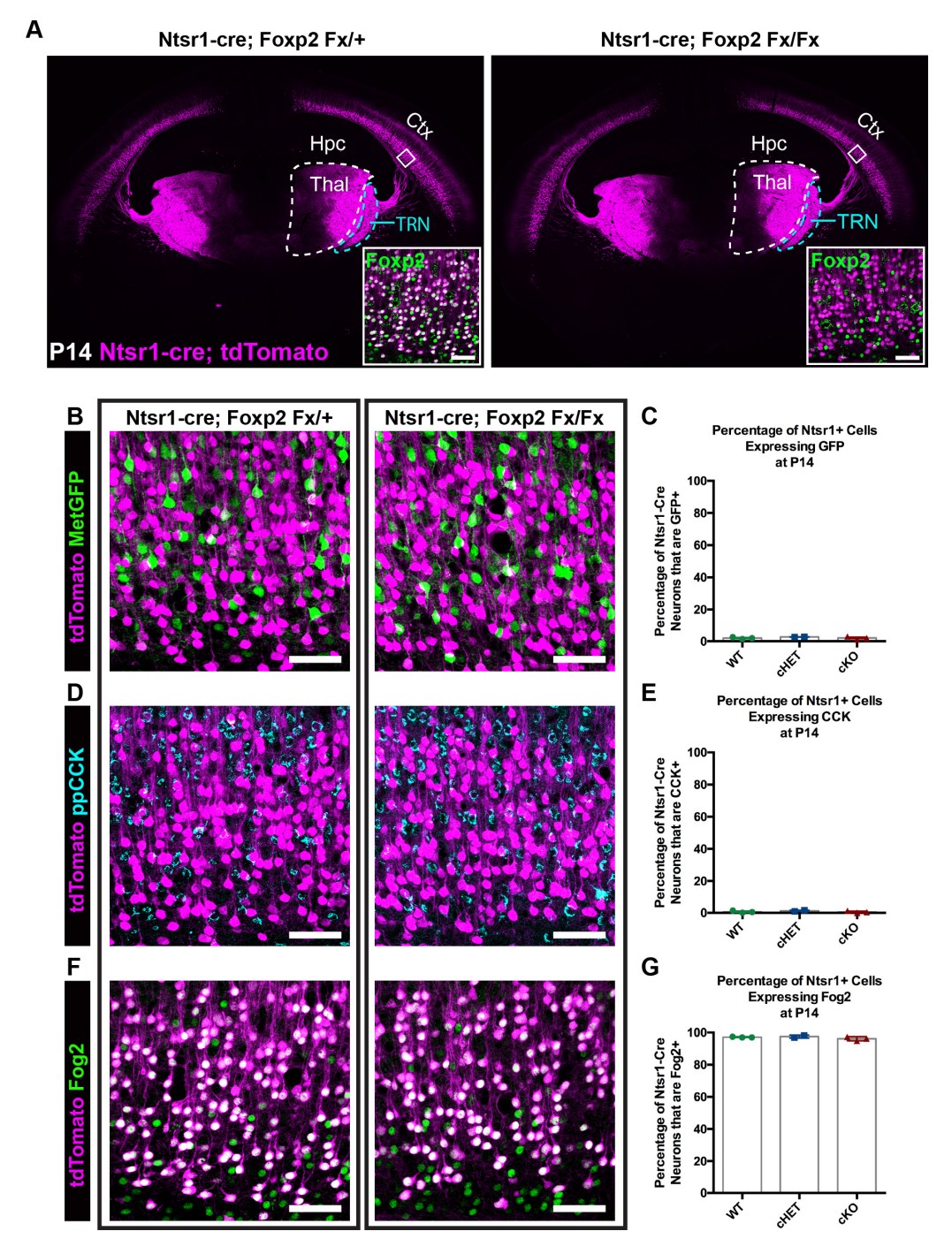

**Figure 2.** FOXP2 is nonessential for class-specific anatomical and molecular phenotypes of corticothalamic neurons. (**A**) At P14 tdTomato (magenta) expression in layer 6 corticothalamic neurons of *Ntsr1-cre; Rosa-tdTomato* mice reveals similar organization of corticothalamic innervation in *Foxp2* conditional knockout mice and heterozygous littermates – boxed inset shows removal of FOXP2 protein (green) from tdTomato$^+$ corticothalamic neurons of *Ntsr1-cre; Foxp2$^{Fx/Fx}$* mouse. (**B**) Met$^{GFP}$ (green) and tdTomato (magenta) label distinct cell populations in *Foxp2* conditional knockout mice, heterozygous littermates, and wild-type C57Bl/6J mice. (**C**) Quantification of co-expression of GFP by tdTomato$^+$ corticothalamic neurons across *Foxp2* genotypes (WT = *Ntsr1 cre; Rosa-tdTomato*, no Flox alleles, N = 3 mice; cHET = *Ntsr1 cre; Foxp2$^{Fx/+}$*, N = 2 mice; cKO = *Foxp2* Fx/Fx, N = 3 mice). (**D**) ppCCK expression is excluded from layer 6 corticothalamic neurons across *Foxp2* genotypes as indicated by the segregation of tdTomato (magenta) and ppCCK (cyan). (**E**) Quantification of co-expression of CCK by tdTomato$^+$ corticothalamic neurons (N for each group same as panel C). (**F**) FOG2 expression by corticothalamic neurons does not require Foxp2, as nearly all tdTomato$^+$ cells express FOG2 (green) across *Foxp2* genotypes. (**G**)
*Figure 2 continued on next page*

*Figure 2 continued*

Quantification of FOG2 coexpression by tdTomato[+] corticothalamic neurons (N for each group same as panel C). All scale bars, 50 μm. Abbreviations: Ctx, cortex; Hpc, hippocampus; Thal, thalamus; TRN, thalamic reticular nucleus.

DOI: https://doi.org/10.7554/eLife.42012.006

The following source data and figure supplement are available for figure 2:

**Source data 1.** Expression of Layer 6 cell-type markers in Ntsr1-cre; Foxp2Fx mice and controls.

DOI: https://doi.org/10.7554/eLife.42012.008

**Figure supplement 1.** Developmental timing of Cre-mediated recombination in Ntsr1-cre and Emx1-cre.

DOI: https://doi.org/10.7554/eLife.42012.007

within the internal capsule and extensive axonal elaboration in the dorsal thalamus and thalamic reticular nuclei of *Foxp2* conditional knockout mice, their heterozygous littermates and *Ntsr1-cre; Rosa-tdTomato* mice on a wild-type C57Bl/6J background. Consistent with the reported cell-type specificity of the *Ntsr1-cre* driver line, no tdTomato-expressing axons in the corpus callosum or cerebral peduncle were evident for any *Foxp2* genotype at the ages examined. This suggests that FOXP2 is dispensable for the typical positioning of CT neurons in layer 6 and the guidance of their axons to reach and ramify in their normal dorsal thalamic targets.

Recent studies have identified neuronal target genes directly regulated by FOXP2, many of which are directly repressed upon FOXP2 binding (*Spiteri et al., 2007*; *Konopka et al., 2009*; *Mukamel et al., 2011*; *Vernes et al., 2011*). Two such genes, *Cck* and *Met*, display largely non-overlapping expression with FOXP2 within layer 6 (*Figure 1H–J*). Given the repressive effect of FOXP2 on *Met* and *Cck* expression (*Spiteri et al., 2007*; *Mukamel et al., 2011*; *Vernes et al., 2011*), the hypothesis that FOXP2 is required to prevent ectopic expression of *Met* and *Cck* among CT neurons was tested. Mice carrying *Ntsr1-cre; Rosa-tdTomato* and *Met*[GFP] reporter alleles were bred with *Foxp2*[fx] mice. The fraction of tdTomato[+] CT neurons that co-expressed GFP was then quantified in *Foxp2* conditional knockout mice, their heterozygous littermates and *Ntsr1-cre; Rosa-tdTomato* mice on a wild-type C57Bl/6J background at P14. Unexpectedly, the percentage of GFP and tdTomato double-positive neurons was minimal and indistinguishable across genotypes (*Figure 2B,C*; one-way ANOVA, p=0.441; *Ntsr1-cre; Foxp2*[+/+] (WT) mean ± SEM = 1.9 ± 0.4; *Ntsr1-cre; Foxp2*[Fx/+] (cHET), mean ± SEM = 2.7 ± 0.1; *Ntsr1-cre; Foxp2*[Fx/Fx] (cKO), mean ± SEM = 2 ± 0.5), indicating that the exclusion of *Met* expression from CT neurons occurs independent of transcriptional regulation by FOXP2. Next, to determine whether FOXP2 is required to repress CCK expression in CT neurons, colocalization of CCK and tdTomato was quantified. Despite abundant CCK expression among layer 6 neurons, there was minimal co-expression of CCK by tdTomato[+] neurons across *Foxp2* genotypes (*Figure 2C,D*; one-way ANOVA, p=0.5016; *Ntsr1-cre; Foxp2*[+/+] (WT) mean ± SEM = 0.6 ± 0.4; *Ntsr1-cre; Foxp2*[Fx/+] (cHET), mean ± SEM = 1.2 ± 0.4; *Ntsr1-cre; Foxp2*[Fx/Fx] (cKO), mean ± SEM = 0.6 ± 0.4), consistent with selective CCK expression by layer 6 CC neurons and exclusion from CT neurons, as previously reported (*Watakabe et al., 2012*; *Kast et al., 2019*). These data indicate that FOXP2 is not required for the exclusion of CCK and *Met* expression from CT neurons. To determine whether molecular markers unique to CT neurons continue to be expressed in their normal pattern in the absence of FOXP2, labeling of FOG2 among tdTomato-expressing CT neurons was assessed in *Foxp2* conditional knockouts and their heterozygous littermates. FOG2 immunolabeling was detected in nearly 100% of CT neurons, independent of *Foxp2* genotype (*Figure 2E,F*; one-way ANOVA, p=0.3163; *Ntsr1-cre; Foxp2*[+/+] (WT) mean ± SEM = 97.1 ± 0.1; *Ntsr1-cre; Foxp2*[Fx/+] (cHET), mean ± SEM = 97.5 ± 0.7; *Ntsr1-cre; Foxp2*[Fx/Fx] (cKO), mean ± SEM = 96.1 ± 0.7).

Because *Ntsr1-cre* is selectively expressed in CT neurons, it is likely Cre expression begins postmitotically, but the timing of the developmental onset of Cre-mediated recombination in the *Ntsr1-cre* mouse line has not been reported. Given the normal development of CT neurons in *Ntsr1-cre; Foxp2*[Fx] conditional knockout mice, we reasoned that the potentially late timing of the developmental deletion of *Foxp2* could have influenced the lack of abnormal CT phenotypes. This possibility was investigated first by determining the onset of Cre-dependent tdTomato expression in *Ntsr1-cre; Rosa-tdTomato* embryos. Coronal sections of the embryonic forebrain were analyzed on embryonic

days (E)14.5, E16.5, and E17.5. Expression of tdTomato was not detected until E17.5, when it was localized in a sparse population of subplate and layer 6 neurons (*Figure 2—figure supplement 1*). Thus, cre-mediated recombination in the *Ntsr1-cre* line does not begin until approximately E17, well after *Foxp2* expression is initiated in the cerebral cortex (*Figure 2—figure supplement 1*; *Ferland et al., 2003*), and at a developmental time point only shortly preceding the initial innervation of the thalamus by descending CT axons (*Grant et al., 2012*; *Torii et al., 2013*). Thus, it is possible that *Foxp2* operates in an earlier developmental window, prior to *Ntsr1-cre* mediated recombination and the subsequent depletion of previously transcribed and translated *Foxp2*/ FOXP2. To determine whether *Foxp2* might play a role earlier in the development of cortical neurons, the *Emx1-cre* driver line was employed, which exhibits cre-mediated recombination in dorsal pallial progenitors beginning at E10.5, when they have just started to proliferate (*Gorski et al., 2002*). Evaluation of *Emx1-cre* embryos confirmed cre-dependent tdTomato expression by E10.5 (*Figure 2—figure supplement 1B*). In situ hybridization revealed the selective removal of the floxed portion of the *Foxp2^Fx* allele, which encodes the DNA-binding domain, at E14.5 (*Figure 3A*), prior to FOXP2 protein production in the dorsal pallium (*Figure 1A,B*). FOXP2 immunohistochemistry performed on sections of the E16.5 forebrain demonstrated the absence of FOXP2 protein in *Emx1-cre*; *Foxp2^Fx* embryos at the earliest timepoint that FOXP2 could be detected in *Foxp2^Fx* embryos (*Figure 3B*, *Figure 3—figure supplement 1B*). Thus, the removal of *Foxp2* prior to the expression of FOXP2 by any dorsal pallial cells, using *Emx1-cre,* provided the opportunity to assess the developmental role of *Foxp2* function from the earliest stages of cortical development and in the subset of layer 5 PT neurons that normally express FOXP2 (*Figure 1—figure supplement 1*), but that do not express *Ntsr1-cre*.

*Foxp2* is reportedly important for cortical neurogenesis and cell migration (*Tsui et al., 2013*; *Garcia-Calero et al., 2016*); these ontogenetic events thus were evaluated in *Emx1-cre; Foxp2^{Fx/Fx}* conditional knockout mice. FOXP2 immunohistochemistry validated the removal of *Foxp2* from the cortex of *Emx1-cre; Foxp2^{Fx/Fx}* mice at E16.5 (*Figure 2—figure supplement 1C,D*), at P0 (*Figure 3A*), as well as P4 and P14 (data not shown). Immunohistochemistry of the layer-specific markers FOG2, CTIP2 (*Figure 3D*) and DARPP-32 (data not shown) at P0 revealed normal patterns of lamination and cell-density in conditional knockout mice. There also were no overt differences in the thickness of layers as revealed by DAPI staining at P0 or P14 (*Figure 3C,J*). In addition, high-level expression of CTIP2 remained restricted to layer 5, and neurons expressing FOG2, a marker of CT neurons that are enriched for *Foxp2*, were found in their normal position, in layer 6 (*Figure 3D*). The numbers of CTIP2$^+$/FOG2$^-$ cells in layer five were not different between genotypes at P0 (*Figure 3E* Kruskal-Wallis test, p=0.). The numbers of FOG2$^+$ cells in layer 6 were comparable across genotypes (*Figure 3F*; Kruskal-Wallis test, p=0.0437, Foxp2Fx, mean ± SEM = 416 ± 28; cHET, mean ± SEM = 440 ± 68; cKO, mean ± SEM = 357 ± 37; Emx1-cre, mean ± SEM = 430 ± 52), as no statistically significant pairwise differences were observed between genotypes (Dunn's Multiple Comparisons test: cKO vs. Foxp2Fx, p=0.3688; cKO vs. Emx1-cre, p=0.1353; cKO vs. cHET, p=0.0674; cHET vs. Foxp2Fx, p=0.9999; cHET vs. Emx1-cre, p=0.9999; Foxp2Fx vs. Emx1-cre, p=0.9999). Importantly, at P14, the numbers of FOG2$^+$ layer 6 CT cells were similar in *Emx1-cre; Foxp2^{Fx/Fx}* and *Foxp2^{Fx/Fx}* littermates (*Figure 3I*, unpaired two-tailed t-test, p=0.6912 *Foxp2^{Fx/Fx}*, mean ± SEM = 209 ± 21; *Emx1-cre; Foxp2^{Fx/Fx}* (cKO), mean ± SEM = 199 ± 13), as were the numbers of CTIP2$^+$/FOG2$^-$ cells in layer 5 (unpaired two-tailed t-test, p=0.99; *Foxp2^{Fx/Fx}*, mean ± SEM = 77 ± 14; *Emx1-cre; Foxp2^{Fx/Fx}* (cKO), mean ± SEM = 79 ± 7). These data demonstrate that the corticofugal populations that express FOXP2 during development do not require FOXP2 for their proper specification. Additionally, the number of ppCCK+ layer 6 CC cells was indistinguishable between conditional knockout and control groups (*Figure 3G,H*. unpaired two-tailed t-test, p=0.5178; *Foxp2^{Fx/Fx}*, mean ± SEM = 139 ± 7; *Emx1-cre; Foxp2^{Fx/Fx}* (cKO), mean ± SEM = 147 ± 9). Thus, the infragranular layers, which contain the neurons that express FOXP2, develop their normal complement of diverse projection neuron subtypes in normal numbers in the absence of *Foxp2*. Finally, the radial thickness of the SSC was indistinguishable across genotypes at P14 (*Figure 3J,K*, unpaired two-tailed t-test, p=0.583), suggesting that FOXP2 function also is dispensable to produce a histogenically and architecturally normal SSC.

Next, to evaluate the putative role of FOXP2 in axon guidance (*Vernes et al., 2011*), *Emx1-cre; Rosa-tdTomato* mice were crossed with *Foxp2^Fx* mice. In agreement with the results from the *Ntsr1-cre; Rosa-tdTomato* experiments, tdTomato-labeled subcortical innervation revealed nearly identical

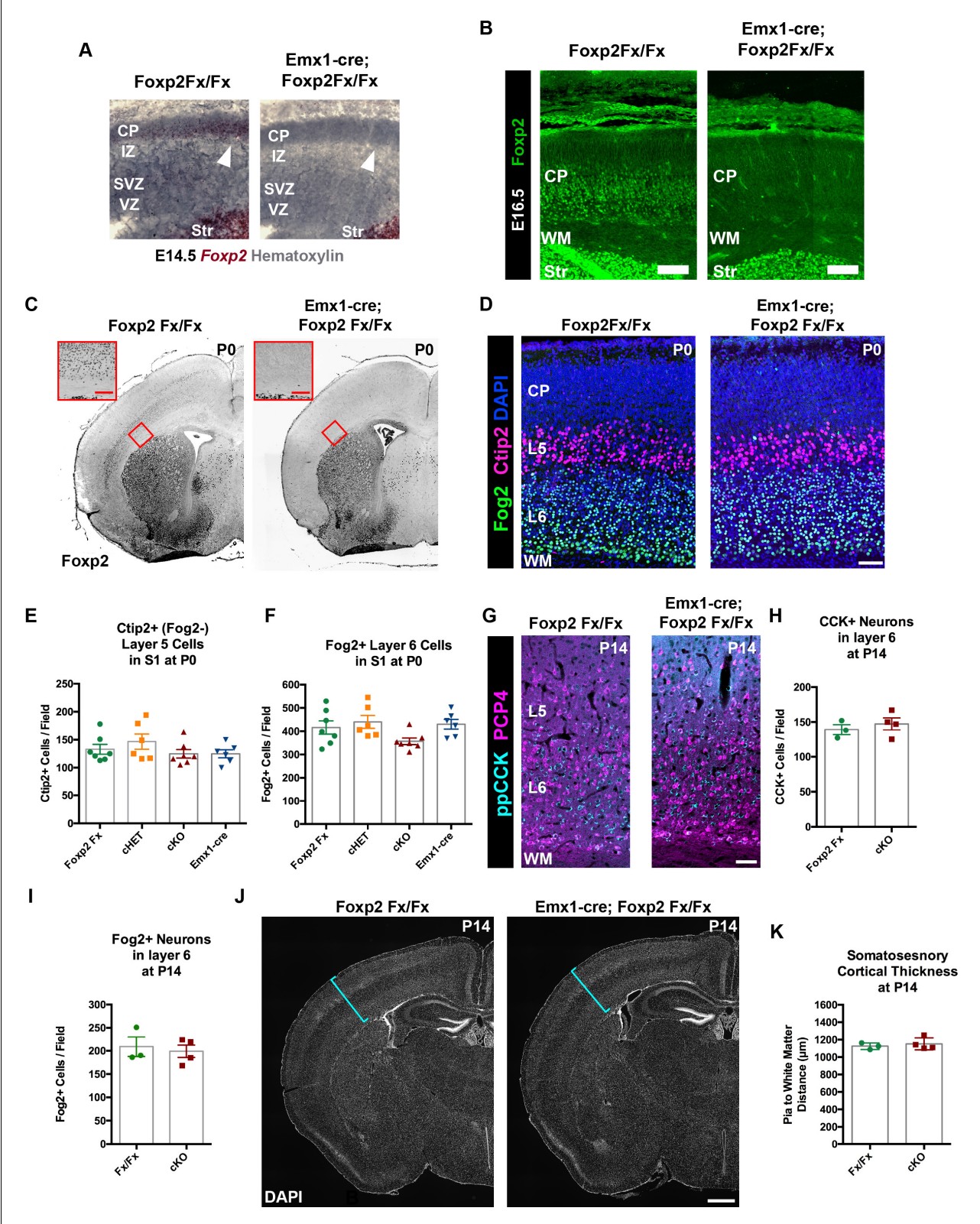

**Figure 3.** FOXP2 is nonessential for the genesis of cortical neurons and their proper lamination. (**A**) *Foxp2 in situ* hybridization based on the BaseScope method reveals expression of *Foxp2* transcript (Red) in *Foxp2^{Fx/Fx}* embryos (N = 4) and selective removal of exons 12–14 (DNA-binding domain) from the dorsal pallium including the cortical plate (white arrowhead) of *Emx1-cre; Foxp2^{Fx/Fx}* mice (N = 6) by E14.5. (**B**) Immunohistochemical analysis of FOXP2 protein in E16.5 embryos (N = 3 each genotype) demonstrates selective elimination of FOXP2 protein (green) from the infragranular layers of

*Figure 3 continued on next page*

*Figure 3 continued*

the dorsal pallium of *Emx1-cre;Foxp2^{Fx/Fx}* mouse embryos. (**C**) FOXP2 immunohistochemistry on coronal sections of P0 *Foxp2^{Fx/Fx}* and *Emx1-cre; Foxp2^{Fx/Fx}* mice reveals absence of FOXP2 (black) in the infragranular cortical layers (red bracket) of *Emx1-cre; Foxp2^{Fx/Fx}* mice.Inset (red outline) shows selective loss of FOXP2 in the cortex at higher magnification. (**D**) FOG2 (green) and CTIP2 (magenta) immunohistochemistry in coronal sections of the primary somatosensory cortex of *Foxp2Fx/Fx* and *Emx1-cre; Foxp2Fx/Fx* mice reveals similar distributions of laminar specific markers at P0. (**E**) Quantification of FOG2^+ cells in layer 6 across genotypes at P0 (Fx/Fx, *Foxp2^{fx/fx}*, N = 7; cHET, *Emx1-cre; Foxp2^{fx/+}*, N = 6; cKO, *Emx1-cre;Foxp2^{fx/fx}*, N = 7; Emx1-cre, N = 6). (**F**) Quantification of CTIP2+/Fog2- cells in layer 5 across genotypes at P0 (N for each group, same as panel E). (**G**) ppCCK (cyan) and PCP4 (magenta) immunohistochemistry in coronal sections of conditional knockout and control littermates at P14. (**H**) Quantification of ppCCK^+ cells in layer 6 of SSC across genotypes at P14 (Foxp2^{Fx/Fx}, N = 3 mice; Emx1-cre; Foxp2^{Fx/Fx}, N = 4 mice). (**I**) Quantification of FOG2^+ cells in layer 6 of SSC across genotypes at P14 (Foxp2^{Fx/Fx}, N = 3 mice; Emx1-cre; Foxp2^{Fx/Fx}, N = 4 mice). (**J**) DAPI-staining of coronal sections of *Foxp2^{Fx/Fx}* and *Emx1-cre; Foxp2^{Fx/Fx}* mice reveals similar size of cortex, including the thickness of primary somatosensory cortex (indicated by cyan bracket). (**K**) Quantification of somatosensory cortex thickness across genotypes at P14 (Foxp2^{Fx/Fx}, N = 3 mice; Emx1-cre; Foxp2^{Fx/Fx}, N = 4 mice). Scale Bars: A (inset), 100 µm; B, E, 50 µm; H, 500 µm.

DOI: https://doi.org/10.7554/eLife.42012.009

The following source data and figure supplement are available for figure 3:

**Source data 1.** Quantification of cortical cell type numbers in Emx1-cre; Foxp2Fx mice and controls.

DOI: https://doi.org/10.7554/eLife.42012.011

**Figure supplement 1.** Elimination of *Foxp2* transcript and protein from the forebrain of *Emx1-cre; Foxp2^{Fx/Fx}* embryos.

DOI: https://doi.org/10.7554/eLife.42012.010

patterns in the internal capsule, thalamus, cerebral peduncle, and pyramidal decussation across *Foxp2* genotypes at P0 (data not shown) and P4 (*Figure 4*). This result, using a genetic deletion strategy prior to cortical neurogenesis, indicates that *Foxp2* is dispensable for proper specification of the cortical neuron subtypes that normally express FOXP2 in SSC, and the appropriate guidance and targeting of their efferent axons.

## Discussion

The current study focused on putative roles for FOXP2 in murine cortical histogenesis, using conditional mouse genetics. The results demonstrate that *Foxp2* is not required for establishing basic developmental organization, molecular phenotypes or efferent connectivity of *Foxp2*-expressing neurons in SSC of mice. The role of FOXP2 in neural development has been of significant interest following the identification of *FOXP2* mutations that cause developmental apraxia of speech in humans (*Lai et al., 2001*; *MacDermot et al., 2005*). Diverse methods and genetic models have been used to interrogate FOXP2 function in a variety of brain areas and species (*French and Fisher, 2014*). Notably, recent studies primarily in mice have implicated *Foxp2* in many developmental processes including cortical neurogenesis (*Tsui et al., 2013*), neuronal migration (*Garcia-Calero et al., 2016*), neuron subtype specification (*Chiu et al., 2014*), neural tissue patterning (*Ebisu et al., 2017*), neurite outgrowth and axon guidance (*Vernes et al., 2011*), synapse formation (*Chen et al., 2016*), and synaptic plasticity (*Groszer et al., 2008*). However, assessment of *Foxp2* function during cerebral cortical development by means of conditional mouse genetics had not been thoroughly pursued. Here, *Foxp2* was deleted at different prenatal ages using dorsal pallial- and cell-type specific Cre-recombinase mouse lines.

### FOXP2 and cortical projection neuron phenotypes

FOXP2 expression is enriched in the deepest layers of the developing and mature neocortex of mammalian species ranging from mice to humans (*Ferland et al., 2003*; *Campbell et al., 2009*; *Mukamel et al., 2011*), suggesting conservation of the cortical cell types that utilize the gene. The connectivity and molecular phenotyping data generated here demonstrate that, in mice, FOXP2 expression within the infragranular layers of the developing SSC is present in nearly all CT neurons, excluded from most layer 6 CC neurons, and transiently expressed by a minor subset of PT neurons. This is consistent with observations of enrichment of *Foxp2* in CT neurons in the primary visual cortex of adult mice by single cell RNA-sequencing and immunohistochemistry (*Tasic et al., 2016*;

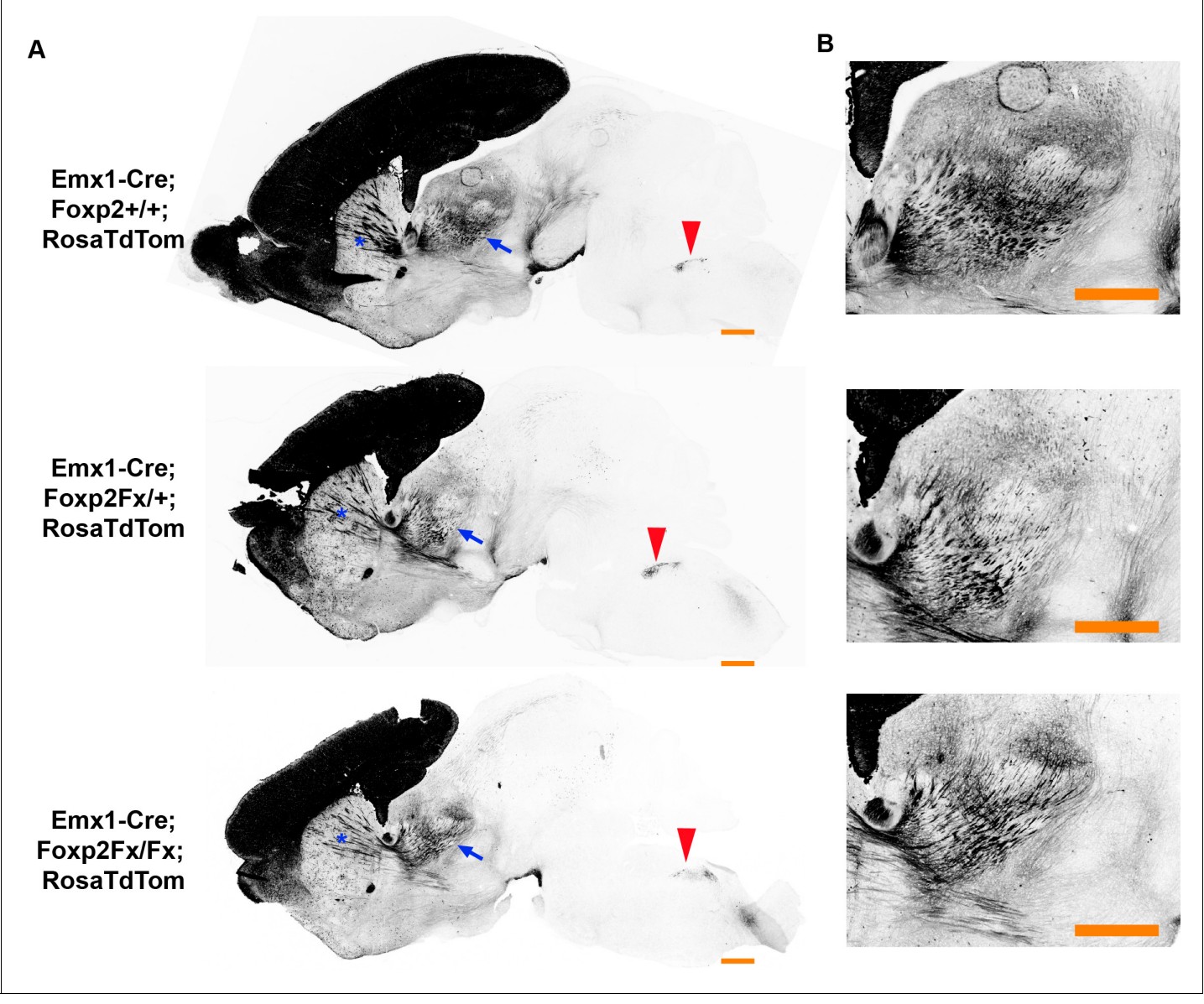

**Figure 4.** FOXP2 is not required for proper corticofugal axon pathfinding. (A) tdTomato reporter (black) reveals similar patterns of corticofugal axon growth in sagittal sections of *Emx1-cre;Foxp2*$^{+/+}$ (WT, top panel, N = 4) *Emx1-cre; Foxp2*$^{Fx/+}$ (middle, N = 3) and *Emx1-cre; Foxp2*$^{Fx/Fx}$ (bottom, N = 3). Note the fasciculation of axons in the internal capsule (blue asterisks) and the comparable growth of axons into the thalamus (blue arrows) and pyramidal decussation (red arrowheads). (B) Higher magnification images of the corticothalamic innervation patterns in each genotype. Scale Bars = 500 μm.

DOI: https://doi.org/10.7554/eLife.42012.012

*Sundberg et al., 2018*), and previous findings that FOXP2 is expressed by subsets of layer five neurons (*Ferland et al., 2003*; *Hisaoka et al., 2010*; *Molyneaux et al., 2015*), which we show here are PT-type neurons. Analysis of SSC at three postnatal ages demonstrated that expression of FOXP2 by CT-type neurons is stable, whereas PT-type neurons express FOXP2 transiently, with no detectable FOXP2 expression in most PT-type neurons at P14. The developmental cell-type selectivity of FOXP2 expression raises important questions regarding the function of FOXP2.

With the enrichment of FOXP2 in CT neurons, and very limited expression in CC neurons, the current study addressed whether *Foxp2* expression is required to repress expression of two putative

target genes, *Cck* and *Met*, which are generally excluded from CT neurons. Such a role would be consistent with the non-overlapping expression patterns of these genes with *Foxp2* in layer 6, as well as previous reports of direct repressive control by FOXP2 in vitro (*Spiteri et al., 2007*; *Mukamel et al., 2011*; *Vernes et al., 2011*). However, deletion of *Foxp2* failed to alter expression patterns of *Met* or *Cck*, suggesting other transcriptional mechanisms may mediate their cell type-specific expression in infragranular layers in vivo. Additionally, two important molecular features unique to FOXP2$^+$ neurons, FOG2 and DARPP-32 expression, are unchanged following either very early or late prenatal deletion of *Foxp2*. Much broader molecular profiling is warranted, but the data indicate that *Foxp2* is neither required for the specification of some of the unique molecular features of FOXP2$^+$ layer 6 CT neurons, nor for the regulation of alternate cell-type molecular signatures that were predicted from previous analysis of FOXP2 transcriptional regulatory targets.

## FOXP2 and early cortical neuron development

Using the early expressing *Emx1-cre* driver line (E10.5), the data show that *Foxp2* deletion does not disrupt the normal generation and migration of neurons in SSC. Thus, unlike other transcriptional regulators of cell-type identity (e.g. CTIP2, FEZF2, TBR1), for which dramatic changes in cell type numbers and projection phenotypes develop upon mutation (*Arlotta et al., 2005*; *Chen et al., 2005*; *Han et al., 2011*; *McKenna et al., 2011*; *Molyneaux et al., 2005*), FOXP2 appears dispensable for the general production of cortical neurons and the specification of the specific projection populations that normally express FOXP2. This conclusion is consistent with the absence of developmental defects recently reported in *Nex1-cre; Foxp2$^{Fx}$* mice, in which *Foxp2* is deleted from postmitotic excitatory neurons of the dorsal pallium (*Medvedeva et al., 2018*). These results were unexpected given the data in previous studies using in utero electroporation (IUEP)-mediated *Foxp2* shRNA knockdown, which demonstrated atypical cortical neurogenesis and migration (*Tsui et al., 2013*; *Garcia-Calero et al., 2016*). In fact, using IUEP and identical shRNA reagents, we observed similar phenotypes to those previously reported (data not shown) (*Tsui et al., 2013*). Several explanations, based on distinct methodologies, could account for the discrepant findings using IUEP knockdown compared to hetero- and homozygous genetic deletion. Recent studies have revealed compensatory transcriptional responses by orthologous transcripts in some mutant mouse lines, and thus it is possible that other Foxp family members could compensate for Foxp2 removal in our studies (discussed more below). Additionally, when crossed with the *Foxp2$^{Fx}$* conditional allele, *Emx1-cre* leads to uniform removal of *Foxp2* from dorsal pallial progenitors, whereas in utero electroporation reduces gene expression in a much smaller subpopulation of cells. This generates a mosaic of *Foxp2*-positive and negative cells. The altered neurogenesis phenotype observed following IUEP mediated-knockdown could result from the removal of *Foxp2* in a mosaic fashion in the dorsal pallium resulting in atypical interactions between neighboring FOXP2$^+$ and FOXP2$^-$ cells. A similar mosaic effect could explain the altered migration observed following IUEP of *Foxp2* shRNA (*Tsui et al., 2013*). Mosaic expression of mutant and wild-type alleles can influence cortical development, shown recently in mouse models of X-linked *Pcdh19* epilepsy (*Pederick et al., 2018*). Resolving whether mosaic *Foxp2* expression can disrupt neurogenesis and migration will require many additional studies using methods distinct from shRNA knockdown, such as in utero electroporation of Cre-expressing plasmid constructs into *Foxp2$^{Fx/Fx}$* embryos. We note, however, the importance of determining the relevance of such mosaic effects, may depend on the identification of a natural context, in humans or developing mouse models, in which mosaic *Foxp2* function occurs in the cortex.

An alternative explanation is that the *Foxp2*-targeting shRNA used in ours and previous studies may lead to incomplete reduction of *Foxp2* expression. This would in turn result in different adaptive responses compared to complete deletion of *Foxp2* genetically. Similar mechanistic explanations have been posed for the discrepant observations of germline versus IUEP-mediated manipulation of doublecortin (*Bai et al., 2003*). Noteworthy is the finding that ectopic overexpression of *Foxp2* results in a paradoxically similar arrest in the radial migration of cortical neurons caused by *Foxp2* shRNA knockdown (*Clovis et al., 2012*). While no direct evidence currently exists, non-physiological mosaic reduction or overexpression of FOXP2 could create an imbalance that itself disrupts cortical development, while genetic disruptions fail to produce the same phenotypes. Finally, shRNA knockdown does have the technical caveat of potential 'off-target' effects, which cannot be ruled out unequivocally. For example, it is possible that the Foxp2 shRNAs also impact the expression of other genes including other Foxp family members, such as Foxp1, which are expressed in the cortex and

could hypothetically compensate for Foxp2 function in our knockout studies. However, in vitro analysis of the specificity of the shRNA constructs suggested that they did not alter the levels of Foxp1 or Foxp4 expression (*Tsui et al., 2013*). Additionally, it is noteworthy that, in a previous study, quantification of *Foxp1* and *Foxp4* transcript levels demonstrated that these genes were expressed at similar levels in WT and *Foxp2* conditional knockout embryos (*French et al., 2007*). Thus, recombination of the *Foxp2^{Fx}* allele does not necesarrily trigger genetic compensation by other Foxp family members. Although unidentified compensatory mechanisms may mask a dispensable role that Foxp2 plays in the core aspects of cortical development studied here, based on results obtained following complete genetic removal of *Foxp2* function, we conclude that *Foxp2* does not play an essential role in murine SSC neurogenesis, neuron migration, subtype specification or axonal pathfinding, contrary to conclusions of other studies.

## Functional implications of FOXP2 in developmental disorders

The lack of overt changes in the generation, migration, differentiation, or axon pathfinding of SSC neurons following conditional *Foxp2* deletion, using two different Cre driver lines, is important to consider in several contexts. The results may have implications for understanding the involvement of the cerebral cortex in speech and language impairments associated with *FOXP2* mutations in humans (*Vargha-Khadem et al., 2005*). The results suggest that loss of *FOXP2* function likely does not contribute to these deficits through altered cortical histogenesis. However, it is important to note that the present studies were carried out in mice, and thus it remains possible that novel species-specific roles for *Foxp2* may have been acquired in the human lineage. Nonetheless, the results provide foundational knowledge that will be essential when designing studies to further address the role of FOXP2 in the development and function of specific neuronal cell types in the cortex.

The normal development of cortical phenotypes in *Foxp2* conditional knockout mice is consistent with conventional magnetic resonance imaging of brain structure in patients with *FOXP2* mutations, which did not identify substantive alterations in gray and white matter structure of the cerebral cortex (*Vargha-Khadem et al., 2005*). However, more refined analysis using voxel-based morphometric analysis identified spatially restricted, minor alterations in the gray matter in perisylvian cortical areas of patients with *FOXP2* mutations (*Belton et al., 2003*). Area-restricted deficits in the development of cerebral cortical anatomy may occur in *Foxp2* conditional knockout mice, but were not detected in the present study due to the focus of the analysis on primary SSC. More expansive studies will need to be pursued. In addition, given the lack of speech and language homologous regions in the cerebral cortex of mice, discovery of regional disruptions relevant to humans with *FOXP2* mutations may not be possible.

Here, the genetic deletion of *Foxp2* in mice is distinct from the most common mutations observed in human patients with inherited speech and language abnormalities (*Lai et al., 2001*; *MacDermot et al., 2005*). Cre-mediated recombination of the conditional *Foxp2* allele produces a nonsense mutation that eliminates DNA-binding ability and causes near complete loss of FOXP2 protein (*French et al., 2007*). Functionally analogous, truncating *FOXP2* mutations have been identified in some patients with speech and language disorders (*MacDermot et al., 2005*). However, missense mutations like the one in the KE pedigree are more common and are the best characterized in terms of their associated brain abnormalities (*Vargha-Khadem et al., 1998*; *Lai et al., 2001*). Thus, the conditional knockout mice used here may not fully recapitulate aberrant FOXP2 functions caused by single amino acid changes, which are proposed to elicit dominant-negative functions that could be distinct from the simple loss-of-function caused by conditional *Foxp2* deletion (*Tsui et al., 2013*). For example, missense *FOXP2* mutations could lead to gain-of-function impairments by influencing the activity of other transcription factors. Importantly, transgenic mouse models that carry the same missense mutations as those observed in human populations have been generated and functionally, but not developmentally, characterized (*French and Fisher, 2014*). Irrespective of the differences in the genetic strategies used to disrupt *Foxp2*, the present results strongly suggest that the histogenesis of murine SSC does not depend on transcriptional regulation by FOXP2.

Finally, in *Foxp2* constitutive knockout mice, medium spiny neurons of the striatum display decreased mEPSC frequency, decreased dendritic spine density, and increased mEPSC amplitudes, whereas the macro-level organization and cell-type composition of the striatum remains intact (*Chen et al., 2016*). It would be interesting to investigate whether the cortical neurons that express Foxp2 display similar synaptic abnormalities in *Foxp2* knockout mice. Altered excitability of CT

neurons could contribute to atypical activity within cortico-striato-thalamocortical loops that are important for motor control and information processing, which could significantly alter speech related functions. The demonstration that the histological organization of the somatosensory cortex is unaffected by the removal of *Foxp2* warrants more detailed characterization of cortical circuit function in *Foxp2* mutant mice.

# Materials and methods

## Key resources table

| Reagent type (species) or resource | Designation | Source or reference | Identifiers | Additional information |
|---|---|---|---|---|
| Genetic reagent (*M. musculus*) | *Foxp2Fx (Foxp2tm1.1Sfis)* | *French et al., 2007* | MGI Cat# 3800702, RRID:MGI:3800702 | |
| Genetic reagent (*M. musculus*) | *Rosa-TdTomato (Ai14)* | Jackson Laboratory | IMSR Cat# JAX:007914, RRID:IMSR_JAX:007914 | |
| Genetic reagent (*M. musculus*) | *Ntsr1-cre (GN220)* | MMRRC | MMRRC Cat# 030648-UCD, RRID:MMRRC_030648-UCD | |
| Genetic reagent (*M. musculus*) | *Emx1-cre* | *Gorski et al., 2002* | IMSR Cat# JAX:005628, RRID:IMSR_JAX:005628 | |
| Genetic reagent (*M. musculus*) | MetEGFP BAC (MetGFP) | *Gong et al., 2003* | MGI:6144427 | |
| Antibody | Goat anti-Foxp2 (polyclonal) | Santa Cruz Biotechnology | Cat# sc-21069, RRID:AB_2107124 | IHC (1:100) |
| Antibody | Chicken anti-GFP (polyclonal) | Abcam | Cat# ab13970, RRID:AB_300798 | IHC (1:500) |
| Antibody | Rat anti-Ctip2 (monoclonal) | Abcam | Cat# ab18465, RRID:AB_2064130 | IHC (1:500) |
| Antibody | Guinea Pig anti-ppCCK | *Watakabe et al., 2012* | Dr. Takeshi Kaneko (University of Tokyo) | IHC (1:500) |
| Antibody | Rabbit anti-PCP4 (PEP-19) | Dr. James Morgan (St. Jude's Research Hospital) | | IHC (1:3000) |
| Antibody | Rabbit anti-Fog2 (polyclonal) | Santa Cruz Biotechnology | Cat# sc-10755, RRID:AB_2218978 | IHC (1:250) |
| Antibody | Rabbit anti-DARPP-32 (monoclonal) | Cell Signaling Technology | Cat# 2306, RRID:AB_823479 | IHC (1:500) |
| Antibody | AlexaFluor F(AB')2 488- or 594- or 647- secondaries | Jackson Immunoresearch Laboratories, Inc | | IHC (1:500) |
| Peptide, recombinant protein | Cholera Toxin Subunit B (CTB) | Invitrogen | Cat. #: C-34776, C-34778 | Alexa Fluor Conjugate (555, 647) |
| Commercial assay or kit | BaseScope assay | Advanced Cell Diagnostics | Cat. #: 323971 | |
| Software, algorithm | IMARIS | Imaris (http://www.bitplane.com/imaris/imaris) | RRID:SCR_007370 | Microscopy Image Analysis Software |
| Software, algorithm | Adobe Photoshop | Adobe Photoshop (https://www.adobe.com/products/photoshop.html) | RRID:SCR_014199 | |
| Software, algorithm | Adobe Illustrator (CS6) | Adobe Illustrator (http://www.adobe.com/products/illustrator.html) | RRID:SCR_010279 | |

*Continued on next page*

*Continued*

| Reagent type (species) or resource | Designation | Source or reference | Identifiers | Additional information |
|---|---|---|---|---|
| Software, algorithm | Code used for nuclear immunofluorescent quantification | This paper | | custom written for ImageJ macros (*Source Code 1*; *Source Code 2*) |
| Software, algorithm | ImageJ | ImageJ (http://imagej.nih.gov/ij/) | RRID:SCR_003070 | |
| Software, algorithm | GraphPad Prism 6 | GraphPad Prism (https://graphpad.com) | RRID:SCR_015807 | Version 6 |

## Animals

All animal procedures used in this study were in accordance with the guidelines of the Institutional Animal Care and Use Committee at Children's Hospital Los Angeles. Mice were housed on a 13:11 hr light-dark cycle, and were provided with food and water *ad libitum*. Mice harboring the conditional *Foxp2* allele (*Foxp2$^{Fx}$*; *French et al., 2007*), *Rosa-TdTomato* allele (Ai14), the *Ntsr1-Cre* transgene (GN220), or the *Emx1-Cre* transgene (B6.129S2-Emx1tm1(cre)Krj/J; obtained from Jackson Laboratories) were maintained on an isogenic C57Bl/6J background. MetEGFP BAC transgenic (Met$^{GFP}$) mice were re-derived on the FVB background using the BX139 BAC from the GENSAT collection (*Gong et al., 2003*). Founder mice were backcrossed to C57Bl/6J for at least two generations prior to experimental breeding, such that experiments involving Met$^{GFP}$ mice were carried out on a mixed C57Bl/6J x FVB background. Emx1-Cre first exhibits recombination of floxed loci at embryonic day (E) 10 in mice (*Gorski et al., 2002*), as confirmed in the present study. Based on data reported here, Ntsr1-Cre exhibits recombination initially at E17.

## Retrograde tracing

On postnatal day (P) 12, mice were anesthetized with vaporized isoflurane (5% induction, 1.5–2% maintenance) and stabilized in a Narishige SG-4N small animal head holder. Mice were maintained at 37 °C for the duration of the surgical procedure through a TCAT-2 temperature control device (Physitemp Intruments, Inc) and respiratory rate was continuously monitored to assess depth of anesthesia. Through stereotaxic guidance, a picospritzer connected to a pulled borosilicate pipette (28 μm tip diameter) was used to inject 50–100 nl of Cholera Toxin Subunit B, Alexa Fluor Conjugate (Invitrogen) into the desired cortical or subcortical target. To minimize contamination of unintended brain regions along the needle tract, the pipette was left in place for 5 min before being slowly retracted. Mice received a subcutaneous injection of the non-steroidal anti-inflammatory drug (NSAID) ketoprofen (5 mg/kg) immediately before the surgery and provided with ibuprofen (0.2 mg/mL) in the drinking water until the end of the experiment. After 2 days of recovery, on P14, mice were transcardially perfused with 4% paraformaldehyde dissolved in 1X phosphate buffered saline (PBS) and tissue was processed for immunohistochemical analysis as described below. Stereotaxic coordinates used for P12 mice are as follows: ventrobasal thalamus, AP −1.7, ML 1.3 mm, Depth 3.15 mm; primary motor cortex, AP 0.25 mm, 1.5 mm, 0.9 mm; cerebral peduncle, AP −3.5 mm, 1 mm, 4.8 mm.

## In situ hybridization

The BaseScope assay (Advanced Cell Diagnostics) was performed on embryonic brain sections prepared from *Foxp2$^{Fx/Fx}$* and *Emx1-cre; Foxp2$^{Fx/Fx}$* embryos that were harvested at embryonic day (E) 14.5, with noon on the day of vaginal plug (identified following overnight mating) designated as E0.5. Briefly, the pregnant dam was deeply anesthetized with saturated isoflurane vapor, and cervical dislocation ensured euthanasia prior to embryo dissection. Embryos were decapitated and the brains were submerged in optimal cutting temperature (OCT, Tissue-Tek) compound and frozen in a pre-chilled dry ice and isopropanol slurry, and subsequently stored at −80 °C until cryosectioning. 20 μm coronal cryosections were collected onto SuperFrost Plus (Fisher Scientific) microscope slides, and then stored at −80 °C until in situ hybrization procedures. Slide-mounted sections were removed from −80 °C and immediately fixed by submerging in prechilled 4% PFA in 1X DEP-C PBS for 30 min

on ice with gentle agitation. Sections were dehydrated in 50%, 70%, and two 100% EtOH washes at room temperature for 5 min each. Tissue was stored in 100% EtOH overnight at −20℃. Sections were pretreated with RNAscope Hydrogen Peroxide for 10 min at room temperature and then with RNAscope Protease IV for 15 min at room temperature. Tissue was washed in 1X DEP-C PBS at room temperature. A custom *Foxp2* BaseScope probe (that hybridizes to the floxed region, bases 1832–1977 of *Foxp2* transcript variant 2 (Refseq ID NM_212435.1), of the *Foxp2*$^{Fx}$ allele) was hybridized for 2 hr at 40℃. Sections were washed twice with 1X wash buffer. Amplification and signal detection steps followed the protocol provided in the BaseScope users manual. Slides were counterstained in 25% Hematoxylin solution modified according to Gill III for 2 min at room temperature. Slides were washed in H2O, and then in 0.02% ammonia for 15 s, and H2O once more. Slides were then incubated at 55℃ for 15 min and then mounted in VectaMount (Vector Laboratories). Slides were stored at −20 ℃ until imaged with aLeica DFC295 color camera using brightfield microscopy through a 20x objective lens.

## Immunohistochemistry

Somatosensory cortex (SSC) was the focus of all data analyses reported in the present study. Embryos were decapitated and brains were either immediately submerged in OCT and frozen in a pre-chilled dry ice and isopropanol slurry ('fresh frozen') and stored at −80 ℃ until cryosectioning, or brains were transferred to 4% paraformaldehyde dissolved in PBS (pH 7.4) and incubated at 4 ℃ for 12–18 hr. Early postnatal (P0) mouse brains were dissected in room temperature PBS, transferred to 4% paraformaldehyde dissolved in PBS (pH 7.4) and incubated at 4 ℃ for 12–18 hr. Mice aged P4 or older were perfused transcardially with 4% paraformaldehyde dissolved in PBS. Following perfusion, brains were immediately removed, transferred to 4% paraformaldehyde and incubated at 4 ℃ for 12–18 hr. Following overnight fixation, brains were incubated sequentially in 10%, 20% and 30% sucrose dissolved in PBS for 12–24 hr each. Next, brains were embedded in Clear Frozen Section Compound (VWR International) and placed on a weigh boat floating in liquid nitrogen. Once frozen, embedded brains were stored at −80 ℃ until cryosectioning. Twenty μm coronal or sagittal cryosections were cut and collected on SuperFrost Plus slides (Fisher Scientific) at −20 ℃ (PFA fixed tissue) or −15 ℃ (fresh frozen), and then stored at −80 ℃ until immunohistochemical analysis. Before immunostaining, fresh frozen sections were thawed to room temperature, fixed in 4% paraformaldehyde dissolved in PBS at room temperature with agitation for 1 hr and 40 min, and then washed in PBS three times for five minutes each. For immunostaining, sections were warmed at room temperature for 10 min, dried in a hybridization oven at 55 ℃ for 15 min, and then incubated in PBS for 10 min. Blocking and permeabilization were done by incubating sections in PBS containing 5% normal donkey serum and 0.3% Triton X-100 for 1 hr at room temperature. Sections were incubated subsequently in primary antibodies diluted in 0.1% Triton X-100 in PBS overnight at room temperature. Sections were washed five times for five minutes each with 0.2% Tween 20 in PBS. Sections were incubated in Alexa Fluor conjugated secondary antibodies (1:500) diluted in 0.1% Triton X-100 in PBS for 1 hr at room temperature. Sections were washed three times for five minutes each with 0.2% Tween 20 in PBS. Sections were then incubated in 950 nM DAPI in PBS for 8 min, and then subjected to two additional five minute PBS washes. Sections were mounted in Prolong Gold antifade reagent (Life Technologies), and the mounting medium cured for at least 24 hr before collecting confocal microscopy images. Primary antibodies used were as follows: Goat anti-Foxp2 (1:100; Santa Cruz sc-21069), Chicken anti-GFP (1:500; Abcam #ab13970), Rat anti-Ctip2 (1:500; Abcam # ab18465), Guinea Pig anti-ppCCK (1:500; T. Kaneko Lab), Rabbit anti-PCP4 (1:3000; J. Morgan Lab), Rabbit anti-Fog2 (1:250; Santa Cruz sc-10755), Rabbit anti-DARPP-32 (1:500; Cell Signaling #2306).

## Co-localization analyses

Co-localization analysis in SSC was performed as described previously (*Kast et al., 2019*). Briefly, confocal images were collected through a 20x/0.8NA Plan-APOCHROMAT objective lens mounted on a Zeiss LSM 700 confocal microscope with refractive index correction. Optical sections were collected at 1 A.U. and 2 μm z-steps through the entire thickness of each 20 μm section. Colocalization analysis was performed in three-dimensional renderings of each confocal z-stack using IMARIS software (Bitplane).

## Cortical thickness measurements

Fluorescent images of DAPI-stained sections containing SSC were collected through a 5x objective lens mounted on an Axionplan II upright fluorescent microscope (Zeiss), an Axiocam MRm camera (Zeiss) and Axiovision software 4.1 (Zeiss). The images were opened in ImageJ and three lines separated by $\geq$50 μm were drawn in the posteromedial barrel subfield from the pia to the white matter. The length of the nine lines (3 lines x three images) were averaged to give a value for the radial thickness of SSC for each mouse.

## Cell type quantification

Maximum Z-projections were created from confocal z-stacks and custom written ImageJ macros were run to quantify the number of FOG2 +and CTIP2+/FOG2- nuclei at P0. The numbers of FOG2 + and CTIP2+/FOG2- cells for each animal were averaged from three 300 μm wide fields (each separated rostrocaudally by $\geq$200 μm) of the cortex representing the anterior, middle and posterior portions of SSC. The numbers of CCK+ cells within layer 6 at P14 were manually counted by an observer blind to genotype, as the punctate and discontinuous distribution of the immunofluorescent ppCCK signal prevented accurate automated quantitation. Similarly, numbers of FOG2$^+$ and CTIP2+/FOG2$^-$ nuclei at P14 were manually quantified by an observer blind to genotype, due to challenges in automated detection of the lower level expression at this time point.

## Image adjustments and figure preparation

Figures were prepared using Adobe Photoshop and Adobe Illustrator (CS6). Only linear adjustments (i.e. gamma = 1.0) were made to the contrast and signal levels of fluorescence microscopy images, and were done in an identical manner across genotypes.

## Experimental design and statistics

Numbers of biological replicates (number of animals) for each experiment are included in the figure legends. Numbers of animals in each group were chosen in accordance with group numbers in previous publications reporting differences in murine cortical phenotypes similar to those measured in the current study (*Han et al., 2011*; *Woodworth et al., 2016*). Summary statistics and specific statistical tests used are described in the Results section. Parametric tests were used in some cases, although tests for normality were not possible given the modest sample sizes. Statistical analyses were performed using Prism 6 (GraphPad).

## Acknowledgements

We thank Dr. Simon Fisher and Dr. Catherine French for generously sharing the *Foxp2$^{Fx/Fx}$* mice. We thank Dr. James Morgan and Dr. Takeshi Kaneko for providing anti-PCP4 and anti-CCK antibodies, respectively. We thank Amanda Whipple for assistance with animal husbandry, and Dr. Esteban Fernandez of the Children's Hospital Los Angeles Cellular Imaging Core for assistance with image acquisition and analysis.

## Additional information

### Funding

| Funder | Grant reference number | Author |
| --- | --- | --- |
| National Institute of Mental Health | MH067842 | Ryan J Kast<br>Alexandra L Lanjewar<br>Colton D Smith<br>Pat Levitt |
| Children's Hospital Los Angeles | | Ryan J Kast |
| Simms/Mann Institute and Foundation | | Pat Levitt |
| National Institutes of Health | T32GM113859 | Alexandra L Lanjewar |

The funders had no role in study design, data collection and interpretation, or the decision to submit the work for publication.

## Author contributions
Ryan J Kast, Conceptualization, Data curation, Formal analysis, Supervision, Funding acquisition, Validation, Visualization, Writing—original draft, Writing—review and editing; Alexandra L Lanjewar, Formal analysis, Validation, Investigation, Visualization, Methodology, Writing—review and editing; Colton D Smith, Validation, Visualization, Writing—review and editing; Pat Levitt, Conceptualization, Resources, Data curation, Supervision, Funding acquisition, Project administration, Writing—review and editing

## Author ORCIDs
Ryan J Kast [iD] https://orcid.org/0000-0003-3580-8811
Alexandra L Lanjewar [iD] https://orcid.org/0000-0002-2037-0717
Pat Levitt [iD] https://orcid.org/0000-0002-9717-1695

## Ethics
Animal experimentation: All animal procedures used in this study were in strict accordance with the Guide for the Care and Use of Laboratory Animals of the National Institutes of Health. The protocol was approved by the Institutional Animal Care and Use Committee at Children's Hospital Los Angeles (Protocol #374-15).

## Decision letter and Author response
Decision letter https://doi.org/10.7554/eLife.42012.019
Author response https://doi.org/10.7554/eLife.42012.020

## Additional files

### Supplementary files
• Source code 1. Maximum Projection and Cropping Macro.
DOI: https://doi.org/10.7554/eLife.42012.013
• Source code 2. Cortical Cell Type Autocounting Macro for nuclear markers.
DOI: https://doi.org/10.7554/eLife.42012.014
• Transparent reporting form
DOI: https://doi.org/10.7554/eLife.42012.015

### Data availability
All numbers relating to quantitative experiments have been uploaded to Dryad (https://dx.doi.org/10.5061/dryad.6hd7bf7).

The following dataset was generated:

| Author(s) | Year | Dataset title | Dataset URL | Database and Identifier |
| --- | --- | --- | --- | --- |
| Kast R, Lanjewar A, Smith C, Levitt P | 2018 | Data from: FOXP2 exhibits neuron class specific expression, but is not required for multiple aspects of cortical histogenesis | https://dx.doi.org/10.5061/dryad.6hd7bf7 | Dryad Digital Repository, 10.5061/dryad.6hd7bf7 |

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
