## [Decision Letter]

Thank you for submitting your article "FOXP2 exhibits neuron class specific expression, but is not required for multiple aspects of cortical histogenesis" for consideration by *eLife*. Your article has been reviewed by three peer reviewers, and the evaluation has been overseen by a Reviewing Editor and Huda Zoghbi as the Senior Editor. The following individual involved in review of your submission has agreed to reveal her identity: Myriam Heiman (Reviewer #2).

The reviewers have discussed the reviews with one another and the Reviewing Editor has drafted this decision to help you prepare a revised submission.

Summary:

Interest in *Foxp2* derives from human genetics, where it is linked to developmental language defects. Given the paucity of disease genes linked to language, the original finding of *Foxp2* defects in humans was very exciting. Studies in the mouse suggested that it is important in development of the cerebral cortex, specifically layer 6 cortical neurons. Here, the authors use mouse genetics to delete FoxP2 from the developing mouse cortex. They did not find defects in cortical neuron genesis, migration, or corticothalamic connectivity. Other defects may be present, but were not characterized. These data are important to the field, since language and cognitive development of humans is such an interesting, but difficult, topic to address outside of animal models. As the current manuscript shows convincing evidence (but needing more documentation of *Foxp2* loss) that the previous studies from mouse are incorrect, these data are important to set things right.

Essential revisions:

1) Loss of *Foxp2* needs to be definitively established along with the timing of the loss. RNA ISH, using RNAscope, or the new SABER FISH method, can give quantitative data on RNA level or isoform changes. If IHC of IF are used, controls must be convincing.

2) The numbers of animals used for various experiments needs to be provided, and that those numbers should be >2. For negative findings, it is important that we are not looking at some biological oddities.

3) Assess the expression of *Foxp1* following the KO of *Foxp2*. Again, can be done by ISH. If not ISH, then again any protein detection must be done in a convincing and quantitative manner.

*Reviewer #1:*

This is an important study utilizing mouse genetics to assess the requirement of *Foxp2* in cortical development. The findings are predominantly negative building upon previous loss and gain of function studies, which suggested an important role for *Foxp2* in layer 5/6 cortical development. Surprisingly, the findings from the current study instead suggest that *Foxp2* does not appear to be required for deep layer neuron generation, specification, or axonal pathfinding.

Importantly, I think these negative results would be a valuable contribution to the field and would be received with much interest. The study was well-executed and if a few more experiments were performed, I think it could increase confidence in the results. After completion of additional experiments, it should be accepted for publication.

Overall critiques/questions:

To look at the early specification of CT neurons, the authors utilize Emx1-cre tdTomato reporter. In their conditional FOXP2 deletion, is the total number or overall size of the cortex affected? In the Emx1-cre *Foxp2^fl/fl^* mice, for example, it seems as though the overall brain size is affected, at least transiently during development? Figure 2—figure supplement 1 is a good example of that. I can see that at adult stages, the total number of Fog2 positive cells is comparable, but is there a chance that during development there are differences in the timing of their generation that might affect brain function?

A related concern about experimental design, the authors use *foxp2^fl/fl^* as a control to *foxp2^fl/fl^ emx1-cre*, to report that the number of Fog2 cells in layer 6 are reduced, although not significantly. One issue is that the control is not really an appropriate control.

To investigate the early phenotypes of FOXP2, the authors use Emx1-cre and find early evidence of recombination, but only report loss of protein at E16.5, which seems pretty late in layer 6 differentiation. It would be important to investigate further loss of *Foxp2* protein becomes apparent.

The authors emphasize *Foxp2* mutations in speech deficits and forebrain development, which may have an impact in human cortical neurogenesis. However, the studies were performed in mouse. The text should clarify that this is mouse forebrain development in the Abstract/Discussion and *Foxp2* requirement may be different in different species.

*Foxp2* may be very important for circuit formation and function which may explain the disease phenotypes in humans. What are the effects of loss of *Foxp2* on connectivity? Synapse formation? Functional activity?

Could some differences be identified if you looked earlier in development during embryogenesis? Potentially *Foxp2* loss may affect timing of generation, differentiation, maturation etc. which would provide insight into its developmental role.

Although the authors speculate reasons for previously reported findings on *Foxp2*, it is difficult to reconcile with the current study without a more thorough characterization/description of *Foxp2* activity. Are there other genes known to compensate for *Foxp2*? It would be helpful to have some data reconciling the results of this study with previous findings.

*Reviewer #2:*

Recent studies have suggested that FOXP2, a gene that in human is linked to developmental speech impairments, may have a role in cortical neurogenesis and migration. Although a study in the developing mouse brain by the Walsh group demonstrated that cortical *Foxp2* expression is absent from the ventricular zone and restricted to stages of late neuronal migration or during neuronal differentiation, and MRI studies of brain structure in patients with FOXP2 mutations showed no gross defects of the cerebral cortex, mouse overexpression and shRNA knockdown studies have suggested an important role for *Foxp2* in mouse cortical neurogenesis. To more fully investigate the proposed role of *Foxp2* in cortical histogenesis, Kast et al. have used molecular, neuroanatomical, genetic, and circuit level approaches. The data together convincingly show, with the use of two independent conditional genetic approaches, that murine *Foxp2* is not required for cortical neurogenesis or migration. The results are of great importance to the field to clarify the function of FOXP2.

One major point is that loss of *Foxp2* with the conditional genetic approaches is shown only with the use of a goat *Foxp2* polyclonal antibody. It would be important to present RNAScope (or similar in situ with cell type-specific resolution) and global qPCR assessment to show *Foxp2* mRNA depletion based on both conditional approaches.

*Reviewer #3:*

This interesting paper essentially reports negative results: despite a good deal of previous data that supported a role for FoxP2 as being an important transcriptional regulator of layer 6 cortical neuron development, two different conditional knockout strategies reported here both lack any detectable phenotypes. Looking at neuronal projections (corticothalamic) and protein expression in layer 5 and 6 neurons (by immunofluorescence) in both models, no differences with controls were observed. To control for timing of Cre-mediated recombination, the study was carried out using Cre lines with early (Emx1-Cre) and late (Ntsr1-Cre) expression, and presumably recombination.

Overall, this is a thorough and well-reported study. One potential area of concern is that the paper is completely dependent on using immunofluorescence in sections for FoxP2, using a commercial polyclonal antibody, to confirm the timing and degree of loss of FoxP2 function. In terms of major comments, this is the one area that is essential to address with a technically independent method: confirming that either the mRNA or protein are completely absent from the knockout cortex, and when this is the case.

Related to this, the authors speculate a good deal as to why acute knockdown with shRNA would lead to detectable phenotypes, whereas the genetic knockout does not. Aside from the additional experiment suggested above to confirm that this is a true null, the lack of discussion of possible redundancy with or compensation by FoxP1 is a puzzling omission. FoxP1 expression overlaps with FoxP2 in many regions/times – some comment by the authors in the discussion on whether this may or may not compensate for FoxP2 would be helpful.

The conclusion of the paper lacks balance: the authors do not detect gross phenotypes in cell specification or projections, but there are many other features that are not studied here, including neuronal maturation, dendrite development, synaptogenesis etc. Given that FoxP2 is largely expressed post-mitotically in migratory neurons, the title and main conclusions do not accurately reflect the field.

---

## [Author Response]

Essential revisions:1) Loss of Foxp2 needs to be definitively established along with the timing of the loss. RNA ISH, using RNAscope, or the new SABER FISH method, can give quantitative data on RNA level or isoform changes. If IHC of IF are used, controls must be convincing.

We have generated new in situ hybridization (ISH) data using a new method (BaseScope) that demonstrates the loss of the floxed region of the *Foxp2* transcript (exons 12-14) that encodes the DNA binding domain of *Foxp2*. The loss of the gene region encoding the DNA-binding domain demonstrate that the protein product cannot be functional. The data were generated using *Emx1-cre; Foxp2^Fx/Fx^*and control *Foxp2^Fx/Fx^*littermates on embryonic day (E)14.5, a timepoint when FOXP2 protein cannot be detected by immunohistochemistry in WT animals. Importantly, by E16.5, when FOXP2 protein first can be detected by immunohistochemistry in *Foxp2^Fx/Fx^*controls, there is no FOXP2 protein in the cortex of conditional knockouts.

2) The numbers of animals used for various experiments needs to be provided, and that those numbers should be >2. For negative findings, it is important that we are not looking at some biological oddities.

We have increased sample size (N) for all experiments to achieve at least N=3, and typically greater for most experiments reported.

3) Assess the expression of Foxp1 following the KO of Foxp2. Again, can be done by ISH. If not ISH, then again any protein detection must be done in a convincing and quantitative manner.

We respectfully disagree with this request. Hisaoka et al., 2010, reported using double labeling methods that *Foxp1* (concentrated in layers 2-5) and *Foxp2* (concentrated in layer 6, with some expression in layer 5) are expressed almost exclusively in different neuronal populations in the cerebral cortex. There is minor overlap in deep layer 5. There is, however, substantial overlap of *Foxp1* and *Foxp2* expression in the developing striatum, and yet Vernes et al., 2011 nor French et al., 2007 , did not report *Foxp1* expression changes in striatum or whole brain at E16.5, well after *Foxp2* positive neurons are generated. Given the lack of significant overlap in the cerebral cortex, and the hundreds of other genes identified by Vernes et al. that exhibit changes in expression in mice carrying a *Foxp2* mutation, we believe the rationale is weak for doing further studies of *Foxp1* expression. We do believe it would be of interest to do an in depth profiling of gene expression using single cell RNA sequencing (scRNAseq) of *Foxp2*-expressing layer 6 and 5 neurons, but that profiling would require follow-up functional studies to determine the relevant developmental adaptations, which we argue is well beyond the scope of the current report. Finally, as we note below in response to reviewer #1, changes in gene expression of other Foxp family members, or any other genes, due to *Foxp2* loss, would not provide conclusive mechanistic evidence for the lack of a phenotype, nor would it change the main conclusion of this report that *Foxp2* is not required for cortical development phenotypes that have been assigned previously to *Foxp2*.

Reviewer #1:[…] Overall critiques/questions:To look at the early specification of CT neurons, the authors utilize Emx1-cre tdTomato reporter. In their conditional FOXP2 deletion, is the total number or overall size of the cortex affected? In the Emx1-cre Foxp2^fl/fl^ mice, for example?

The numbers of CT neurons at P0 were quantified in cKO (*Emx1-cre; Foxp2Fx*) and animals of 3 other genotypes (*Emx1-cre; Foxp2^Fx/Fx^* (cHET), Emx1-cre on WT C57Bl6 background, and *Foxp2^Fx^*^/Fx^), which did not contain tdTomato reporter. The results (Figure 3C, D) demonstrated that the number of CT neurons were not statistically significantly different across genotypes. Additionally, the radial thickness of the cortex was measured at P14 and found to be indistinguishable across genotypes (Figure 3H, I).

It seems as though the overall brain size is affected, at least transiently during development. Figure 2—figure supplement 1 is a good example of that.

We thank the reviewer for the observation. We have looked more carefully at this and included comparisons of very closely matched (in terms of anterior-posterior level) coronal sections from cKO and control littermates. The new data show that the size of the brains are not different across genotypes at E14, E16, P0, or P14. We have replaced the images in Figure 2—figure supplement 1 with these new images at E16. These sections also were stained with an optimized immunohistochemical procedure to convincingly show the presence and absence of cortical FOXP2+ cells in the control and cKO brains, respectively.

I can see that at adult stages, the total number of Fog2 positive cells is comparable, but is there a chance that during development there are differences in the timing of their generation that might affect brain function?

We have quantified the numbers of layer 6 neuron subtypes at P0 and P14 and found no differences across genotypes at either age. So, any developmental differences would be limited to the late prenatal period. We have not performed an exhaustive analysis of cell types across genotypes during prenatal development, so we cannot rule out such a change. The hypothesized rate changes of neuronal subtype differentiation would need to be limited to a 72-hour developmental window, between E16.5 (FOXP2 onset) and P0.5 (when we observe no differences in cell numbers). We note that addressing this hypothesis would require analysis at many prenatal time points (to account for variation in developmental timing due to the imprecision in determining fertilization), and thus many more months of experimentation. While possible, we believe rate changes in differentiation are unlikely, and moreover would not explain the end result of normal subpopulation representations and other normal phenotypes in the *Foxp2* cKO.

A related concern about experimental design, the authors use foxp2^fl/fl^ as a control to foxp2^fl/fl^ emx1-cre, to report that the number of Fog2 cells in layer 6 are reduced, although not significantly. One issue is that the control is not really an appropriate control.

We have updated the text to properly emphasize that there is no statistical difference in the numbers of Fog2 cells across genotypes. We share the reviewer’s emphasis on properly controlling for genotypes in these experiments, which is why 4 genotypes were included in experiments that suggested a trending effect when comparing *Emx1-cre; Foxp2^Fx/Fx^*and *Foxp2^Fx/Fx^*littermates (e.g. Fog2 counts at P0). We note that most developmental studies of similar type to the report here do not include 4 relevant genotypes, as we have done. In experiments in which initial comparisons of *Emx1-cre; Foxp2^Fx/Fx^*and *Foxp2^Fx/Fx^* groups did not show any differences (i.e. cell type counts and cortical thickness measurements at P14), we chose not to include a Cre-only control group as there was no effect of *Foxp2* removal. We added the Cre-only control group in cases where initial data indicated that there might be a trend. We have repeated the experiment by increasing the sample sizes (N) of each group in the P0 cell type counting experiment:

1 – *Foxp2^Fx/Fx^* from n=5 to n=7

2 – cHet from n=5 to n=6

3 – cKO from n=4 to n=7

4 – Emx1-cre only from n=4 to n=6

The results of the experiment, and the application of appropriate non-parametric statistics, with correction for multiple comparisons, continue to support our conclusion that the numbers of Fog2 cells are not different across genotypes, and thus *Foxp2* does not influence the specification of this cortical cell type.

To investigate the early phenotypes of FOXP2, the authors use Emx1-cre and find early evidence of recombination, but only report loss of protein at E16.5, which seems pretty late in layer 6 differentiation. It would be important to investigate further loss of Foxp2 protein becomes apparent.

We have added ISH data based on the BaseScope(Advanced Cell Diagnostics) technique, which demonstrates that in *Emx1-cre; Foxp2^Fx/Fx^*mice the floxed *Foxp2* allele undergoes recombination by E14.5, thus leading to the complete absence of the mRNA segment encoding the *Foxp2* DNA-binding domain. This precedes the expression of FOXP2 protein, which we could not detect through IHC until E16.5. We note that our assessment of initial expression of FOXP2 protein is consistent with that reported by Ferland et al., 2003, and in the in situ hybridization data publicly available in the Allen Brain Institute database for developing mice.

The authors emphasize Foxp2 mutations in speech deficits and forebrain development, which may have an impact in human cortical neurogenesis. However, the studies were performed in mouse. The text should clarify that this is mouse forebrain development in the Abstract/Discussion and Foxp2 requirement may be different in different species.

We have updated the text in the Abstract and the Discussion to emphasize that our work has been done in mice and leave open the possibility that there may be specific functions of *Foxp2* related to cortical development in other species.

Foxp2 may be very important for circuit formation and function which may explain the disease phenotypes in humans. What are the effects of loss of Foxp2 on connectivity? Synapse formation? Functional activity?

We agree with this statement. For example, it is clear that disruption of *Foxp2* expression in the striatum has developmental and functional outcomes. However, we wish to re-emphasize that the results of experiments probing later aspects of brain development and function would be difficult to interpret in the absence of the foundational data provided in the present study. Moreover, in the context of the dramatic effects attributed to *Foxp2* in previous publications, such experiments would seem somewhat distant from the primary action of *Foxp2*. Given the lack of overt histological changes, we agree that as a follow-up of the current detailed analysis, it would be of interest to investigate the role of *Foxp2* in later aspects of brain development, plasticity and function. We believe the current study will assist in focusing those in the field on other epochs of development that speak to the issue raised by this reviewer.

Could some differences be identified if you looked earlier in development during embryogenesis? Potentially Foxp2 loss may affect timing of generation, differentiation, maturation etc. which would provide insight into its developmental role.

This issue is related to our third response to reviewer 1 above and we have addressed the challenges of the hypothesis of evaluating timing of maturation in a short window from the onset of *Foxp2* expression to our initial quantitative analysis (72 hr). As noted for #6, we believe follow-up studies are merited, but are beyond the scope of the present study. We have attempted to be very careful about the conclusions drawn with regard to speaking only to the time periods and measures that we have conducted.

Although the authors speculate reasons for previously reported findings on Foxp2, it is difficult to reconcile with the current study without a more thorough characterization/description of Foxp2 activity. Are there other genes known to compensate for Foxp2? It would be helpful to have some data reconciling the results of this study with previous findings.

This is an interesting and valid point. Indeed, pioneering work by Edward Morrisey and colleagues demonstrated the functional cooperativity of Foxp family members including *Foxp1*, 2, and 4 in the development of anterior gut derived structures including the esophagus and airway (Li et al., 2004; Shu et al., 2007). These studies demonstrated that compound *Foxp2*^-/-^ and *Foxp1*^-/+^ mutants display more dramatic lung development defects than *Foxp2*^-/-^ single mutants, but in the case of the lung, even *Foxp2*^-/-^ single mutants showed defects in anterior gut developmental phenotypes. The present neurodevelopmental study aimed to build on previous publications that concluded that *Foxp2* plays a critical role in corticogenesis. However, we found that previous conclusions drawn about the role of *Foxp2* in cortical development (suggested to occur independent of manipulating other Foxp family members using shRNA technology) are inconsistent with the genetic approaches used here, with 2 different Cre driver lines, performed in the present study. Genetic manipulations are typically taken as the gold standard for determining gene function, although other approaches can be valuable. Thus, while the possibility still exists that compensation for the loss of *Foxp2* function [by any mechanism, including *Foxp1* upregulation, or upregulation of the hundreds of transcripts reportedly regulated by *Foxp2* (Vernes et al., 2011)] might mask a valid, but nonessential, role for *Foxp2* in the development of the cortex, the main conclusion drawn here – that *Foxp2* is not required for basic cortical histogenic events that include cell type production in which *Foxp2* is expressed, and fundamental cortical axon extension and targeting to the dorsal thalamus and pyramidal tract – is supported firmly by the present data. Developmental manipulations can also be assessed for potential compensatory mechanisms, because phenotypes could be a sum of direct loss of gene activity coupled with adaptations. These types of studies are of great interest, and will require significant new detailed molecular, developmental and functional experiments that will likely take years, given the number of genes that are regulated putatively by Foxp family members. We believe this is well beyond the scope of the present study, and will not impact the conclusions drawn that *Foxp2* is not required for specific aspects of cortical development in mice.

Reviewer #2:[…] One major point is that loss of Foxp2 with the conditional genetic approaches is shown only with the use of a goat Foxp2 polyclonal antibody. It would be important to present RNAScope (or similar in situ with cell type-specific resolution) and global qPCR assessment to show Foxp2 mRNA depletion based on both conditional approaches.

We have added new ISH data based on a new method (BaseScope) that demonstrates complete absence of *Foxp2* exons 12-14 (floxed region, encodes DNA-binding domain) in the dorsal pallium of *Emx1-cre; Foxp2^Fx/Fx^*mice from at least E14.5 onward due to the recombination of the floxed *Foxp2* allele in the cortex. We note that we were unable to detect FOXP2 protein until E16.5, and thus genetic recombination occurs prior to FOXP2 protein production.

Reviewer #3:[…] One potential area of concern is that the paper is completely dependent on using immunofluorescence in sections for FoxP2, using a commercial polyclonal antibody, to confirm the timing and degree of loss of FoxP2 function. In terms of major comments, this is the one area that is essential to address with a technically independent method: confirming that either the mRNA or protein are completely absent from the knockout cortex, and when this is the case.

As described above, we’ve added new ISH data based on a new method (BaseScope) that demonstrates complete absence of *Foxp2* exons 12-14 (floxed region, encodes DNA-binding domain) in the dorsal pallium of *Emx1-cre; Foxp2^Fx/Fx^*mice. Please see response to reviewer 2.

Related to this, the authors speculate a good deal as to why acute knockdown with shRNA would lead to detectable phenotypes, whereas the genetic knockout does not. Aside from the additional experiment suggested above to confirm that this is a true null, the lack of discussion of possible redundancy with or compensation by FoxP1 is a puzzling omission. FoxP1 expression overlaps with FoxP2 in many regions/times – some comment by the authors in the Discussion on whether this may or may not compensate for FoxP2 would be helpful.

As noted above (Essential revisions, comment #3), *Foxp2* and *Foxp1* are expressed mostly in non-overlapping neuronal populations in the developing and adult cerebral cortex. We have added discussion to address this. Also, as noted, given the plethora of genes with Foxp family member DNA binding sites and that are hypothetically regulated by *Foxp2*, compensation is certainly possible, and we note this.

The conclusion of the paper lacks balance: the authors do not detect gross phenotypes in cell specification or projections, but there are many other features that are not studied here, including neuronal maturation, dendrite development, synaptogenesis etc. Given that FoxP2 is largely expressed post-mitotically in migratory neurons, the title and main conclusions do not accurately reflect the field.

We have edited the manuscript to ensure accuracy in the conclusions drawn from these studies. While limited to the late prenatal and early postnatal periods, we do not believe that the phenotypes we studied are gross in nature. Cell type specification is an essential histogenic event for which connectivity, circuit formation and function all depend upon. We agree that doing studies later in development may be useful for follow-up, but would not change the careful conclusions that we have drawn.